# Hippocampal Noradrenaline Is a Positive Regulator of Spatial Working Memory and Neurogenesis in the Rat

**DOI:** 10.3390/ijms24065613

**Published:** 2023-03-15

**Authors:** Rosario Gulino, Domenico Nunziata, Gioacchino de Leo, Anna Kostenko, Serena Alexa Emmi, Giampiero Leanza

**Affiliations:** 1Department of Biomedical and Biotechnological Sciences, University of Catania, 95123 Catania, Italy; 2B.R.A.I.N. (Basic Research and Integrative Neuroscience) Laboratory for Neurogenesis and Repair, Department of Life Sciences, University of Trieste, 34100 Trieste, Italy; 3Department of Drug and Health Sciences, University of Catania, 95125 Catania, Italy; 4Molecular Preclinical and Translational Imaging Research Centre—IMPRonTE, University of Catania, 95125 Catania, Italy

**Keywords:** Alzheimer’s disease, noradrenaline, locus coeruleus, hippocampus, immunolesion, transplantation, radial arm water maze, progenitor proliferation, rat

## Abstract

Loss of noradrenaline (NA)-rich afferents from the Locus Coeruleus (LC) ascending to the hippocampal formation has been reported to dramatically affect distinct aspects of cognitive function, in addition to reducing the proliferation of neural progenitors in the dentate gyrus. Here, the hypothesis that reinstating hippocampal noradrenergic neurotransmission with transplanted LC-derived neuroblasts would concurrently normalize both cognitive performance and adult hippocampal neurogenesis was investigated. Post-natal day (PD) 4 rats underwent selective immunolesioning of hippocampal noradrenergic afferents followed, 4 days later, by the bilateral intrahippocampal implantation of LC noradrenergic-rich or control cerebellar (CBL) neuroblasts. Starting from 4 weeks and up to about 9 months post-surgery, sensory-motor and spatial navigation abilities were evaluated, followed by post-mortem semiquantitative tissue analyses. All animals in the Control, Lesion, Noradrenergic Transplant and Control CBL Transplant groups exhibited normal sensory-motor function and were equally efficient in the reference memory version of the water maze task. By contrast, working memory abilities were seen to be consistently impaired in the Lesion-only and Control CBL-Transplanted rats, which also exhibited a virtually complete noradrenergic fiber depletion and a significant 62–65% reduction in proliferating 5-bromo-2′deoxyuridine (BrdU)-positive progenitors in the dentate gyrus. Notably, the noradrenergic reinnervation promoted by the grafted LC, but not cerebellar neuroblasts, significantly ameliorated working memory performance and reinstated a fairly normal density of proliferating progenitors. Thus, LC-derived noradrenergic inputs may act as positive regulators of hippocampus-dependent spatial working memory possibly via the concurrent maintenance of normal progenitor proliferation in the dentate gyrus.

## 1. Introduction

The pontine Locus Coeruleus (LC) nucleus is the main source of noradrenaline (NA)-containing fiber innervation for the whole central nervous system and plays a critical role in the regulation of many brain functions, including vigilance, attention, anxiety, depression and cognition [1,2,3,4]. Growing evidence from the past decade (reviewed in [5,6,7,8]) has shown that in aging and Alzheimer’s Disease (AD) patients, this nucleus undergoes profound anatomical and functional alterations closely related to the severity of the cognitive deficits, when present, and may precede the manifestation of clinical symptoms [9,10,11,12,13,14,15]. Notably, in AD patients, a decrease in hippocampal neurogenesis has also been observed to occur at the early stages of the disease, when cognitive impairment is quite mild, suggesting that altered neurogenesis in the hippocampus may indeed precede and perhaps contribute to cognitive decline in AD [16,17]. Thus, investigating the possible relationships between central noradrenergic function and hippocampal neurogenesis in AD is highly relevant for a better understanding of the possible causes underlying, or contributing to, memory decline and cognitive deterioration in the disease (see [18] for review). On the pre-clinical side, several studies have begun to investigate in experimental animals the contribution of the ascending noradrenergic afferents to neural progenitor proliferation in the subgranular zone (SGZ) of the hippocampal dentate gyrus [19,20,21,22,23]. In general, all of these studies have built consensus on viewing the LC noradrenergic projection system as a positive regulator of hippocampal neurogenesis.

Surprisingly, however, not much work has addressed the possible relationships between NA-regulated neurogenesis and hippocampus-dependent cognitive abilities. In fact, in the only published study investigating the effects upon both cognitive function and hippocampal neurogenesis of selective LC lesions in adult animals [22], severe working memory deficits were detected in hippocampus-dependent spatial navigation tasks which correlated with a reduced number of proliferating precursors bilaterally in the SGZ. Thus, it appears highly plausible that the influence exerted by LC-derived noradrenergic inputs to regulate the complex series of events leading to the proliferation of newly generated progenitors in the SGZ may also involve the regulation of distinct aspects of cognitive function (i.e., working memory). Interestingly, reinstating hippocampal noradrenergic neurotransmission using, e.g., locally implanted noradrenergic neuroblasts, has recently been observed to promote a significant amelioration of working memory performance disrupted by the lesion [24,25]. However, no study to date has addressed whether such transplant-promoted recovery of working memory abilities would also affect the proliferation of precursor cells in the hippocampal SGZ.

In the present study, discrete injections of the selective immunotoxin anti-dopamine beta hydroxylase saporin (anti-DBH-saporin) were carried out bilaterally in 4-day-old rats to deplete the noradrenergic innervation of the hippocampus, followed, 4 days later, with the bilateral implantation of LC-derived or cerebellar (CBL) neural progenitors in the denervated hippocampal regions. The aim was to investigate whether the reinstatement of hippocampus-dependent working memory with reinnervating, locally implanted, noradrenergic neuroblasts would also affect the proliferation of precursor cells in the hippocampal SGZ.

## 2. Results

### 2.1. General Observations

All animals, irrespective of their treatment, increased in body weight and exhibited fairly normal sensory-motor functioning when evaluated in both bridge and grid tests at about 5 weeks of age (Table 1).

Thus, no lesion-induced sensory-motor deficits were detected that would affect performance in the navigation tasks (see also below). Rats with vehicle injections did not differ from the intact animals on any of the behavioral or morphological parameters analyzed. These animals were therefore combined into a single Control group (Control, *n* = 12) for all analyses and illustrations.

### 2.2. Behavioural Analyses

#### 2.2.1. Morris Water Maze

Figure 1A,B illustrates the performance of the groups in a cued paradigm of the Morris water maze task, in which the platform was signaled and its position varied on each of the four daily trials. This test was administered to check for possible lesion-induced non-cognitive (e.g., visual) deficits that would affect performance. In general, animals in all groups improved their performance over time (two-way mixed ANOVA, the effect of day on latency, F_2,96_ = 89.72; on distance, F_2,96_ = 80.88; both *p* < 0.001) and did not differ from each other (main group effect on latency F_3,48_ = 0.26; on distance, F_3,48_ = 0.01; group × day on latency, F_6,96_ = 1.09; on distance, F_6,96_ = 2.12; all n.s.).

Mean latencies and swim distances required to find the platform in the place test, are shown in Figure 2A,B. All animals initially required about 38 s and 9 m to locate the hidden platform but improved significantly with repeated training (two-way mixed ANOVA, the effect of day on latency, F_6,288_ = 178.79; on distance, F_6,288_ = 190.28; both *p* < 0.001). In fact, they learned at similar rates throughout the seven days of testing, and the groups did not differ from each other (main group effect on latency, F_3,48_ = 0.39; on distance, F_3,48_ = 0.29; group × day on latency, F_18,288_ = 0.50; on distance, F_18,288_ = 0.93; all n.s.). Swim speed, monitored as a screen of normal motor capacity during the execution of the navigation tasks, was not seen to differ between groups (one-way ANOVA, main group effect F_3,48_ = 0.33; n.s.) and averaged 0.2–0.3 m/s across the days of testing.

During the spatial probe trial on day 7, the platform was removed and the animals were administered a free 60 s swim (Figure 2C,D). All animals swam primarily in the training (SW) quadrant (effect of the quadrant on swim distance, F_3,144_ = 110.46; on annulus crossings, F_3,144_ = 131.54; both *p* < 0.001) and exhibited a marked bias for the original platform site (main group effect on distance, F_3,48_ = 1.64; on annulus crossings, F_3,48_ = 0.88; group × quadrant interaction on distance, F_9,144_ = 1.0; on annulus crossings, F_9,144_ = 1.78: all n.s.). Groups did not differ for the total number of annulus crossings (one-way ANOVA, main group effect F_3,48_ = 0.79; n.s.). Taken together with the actual swim paths (Figure 2E), these findings indicate an equally active and focused search behavior in all animals.

#### 2.2.2. Radial Arm Water Maze

Group performances in the Radial Arm Water Maze (RAWM) test, administered at about 8 months post-transplantation to evaluate working memory abilities, are shown in Figure 3A–E. In this task, the platform is placed on a new arm every day by the experimenters, thus a new search strategy must be developed by the animals to re-locate it within the five trials of each testing day.

All animals had longer latencies and more arm selection errors during the first trial of each day. They then improved over the five trials (two-way mixed ANOVA, the effect of the trial on latency F_4,192_ = 86.81; on errors F_4,192_ = 82.94; both *p* < 0.001). However, the progressive reduction in time and errors to find the platform differed among groups (main group effect on latency, F_3,48_ = 12.02; on errors, F_3,48_ = 3.33, group × trial interaction on latency, F_12,192_ = 2.0; on errors, F_12,192_ = 2.34; all *p* < 0.05), possibly due to the poor performance exhibited by the Lesioned and

CBL-Transplanted animals relocated the platform less efficiently than animals in the Control and LC Transplant groups (Tukey post hoc comparison for both measures *p* < 0.05; Figure 3A,B). Latency and error savings, reflecting the percent improvement between trials 1 and 2, provided a further measure of learning ability in the RAWM task. Thus, rats in the Control and LC Transplant groups reduced by about 50–60% the latency and entry errors required to reach the platform across trials. By contrast, the Lesioned and the CBL-Transplanted animals were not as efficient (one-way ANOVA + Tukey, post hoc test; *p* < 0.05 for both measures vs. Control and LC Transplant groups) and their improvements never exceeded 15–22% (Figure 3C,D). This is also evident in Figure 3E, where the swim paths of representative animals in the groups on the fifth training day are illustrated.

### 2.3. Morphological Analyses

#### 2.3.1. Lesion and Transplant Effects on DBH-Positive Neurons and Fibers

The bilateral infusion of the anti-DBH-saporin immunotoxin into the hippocampal formation of immature (PD4) rats produced, about 9 months later, a robust loss of DBH-immunoreactive neurons in the LC (Figure 4A–D). The neuronal depletion was particularly evident in the dorsal–central portion of the nucleus, whose cells mainly project to rostrally located targets such as the neocortex and hippocampus, whereas relatively few spared neurons were detected in the more caudal–ventral subcoeruleus nucleus, i.e., where noradrenergic projections to the cerebellum and spinal cord originate [26]. As estimated using stereology (Table 2), the noradrenergic neuronal depletion averaged about 75–78% and was consistently observed in all lesioned animals, irrespective of the hippocampal graft (one-way ANOVA followed by Tukey post hoc test; *p* < 0.01 vs. Control). Thus, DBH-immunoreactive fibers detected in the hippocampus of transplanted animals were largely graft-derived and did not result from incomplete lesions.

The hippocampus of lesioned animals was virtually devoid, and only sparse immunoreactive fibers could be detected in the various subfields (compare, e.g., B with A in Figure 5). Notably, surviving grafts were detected in all animals receiving embryonic noradrenergic-rich tissue bilaterally in the hippocampus. These grafts appeared as either small masses of DBH-positive tissue within the hippocampal fissure (Figure 5C, detailed in E) or groups of individual DBH-positive cells with clear neuronal morphology.

Overall, the transplants were seen as viable, with a correct position and a high degree of anatomical integration within the host hippocampus. Likewise, no evident glial proliferation was observed that would alter the host tissue environment or create a physical barrier for the DBH-immunoreactive fibers outgrowing from the implanted neuroblasts.

In fact, all LC-Transplanted animals exhibited dense networks of graft-derived DBH-positive fibers throughout the hippocampus, giving rise to an organotypic innervation whose distribution pattern was often similar to normal (compare, e.g., C with A in Figure 5). By contrast, in the animals receiving control embryonic cerebellar tissue, the implants, although correctly positioned and extending in the hippocampal fissure, did not produce any significant reinnervation in the lesion-depleted hippocampal regions. The occasional presence of individual DBH-immunoreactive neurons or fibers in these grafts (arrows in Figure 5D) was probably related to the accidental inclusion of adjacent brainstem tissue during the dissection of the cerebellar anlages. However, in no case was such sparse fiber outgrowth seen to induce any clear-cut functional effects. Semi-quantitative analyses of the DBH-positive fiber density in the various subfields, i.e., CA1, CA3 and DG are shown in Table 2. Statistical comparisons (two-way ANOVA followed by Tukey post hoc test), confirmed a dramatic ≈62–70% fiber loss in the Lesioned and CBL-Transplanted animals as compared to both the Control and LC-Transplanted animals (*p* < 0.01). Notably, in spite of the expected variabilities, immunoreactive fibers outgrowing from the transplanted tissue reinstated ≈90–93% of the normal innervation density, particularly in the areas adjacent to the implant, with no obvious side differences.

#### 2.3.2. Effects on BrdU-Positive Progenitors in SGZ

Stereological analyses of the effects of lesioning and grafting upon the proliferation of BrdU-immunoreactive progenitor in the SGZ and hilus of the hippocampal dentate gyrus are listed in Table 3. Estimates were obtained using about 5–7 equally spaced coronal sections encompassing the dorsal hippocampus of animals sacrificed 2 h after the BrdU treatment, so as to address the effect of the lesion and transplant upon cell proliferation. In the Lesioned and CBL-Transplanted animals, a significant ≈63% reduction in the density of proliferating BrdU-immunoreactive nuclei was estimated at the border between the granule cell layer and the hilus (*p* < 0.01) as compared to the Control and LC-Transplanted animals. Notably, the numbers of BrdU-immunoreactive nuclei in these latter animals were seen to be not different from those estimated in Controls, particularly in the vicinity of the grafts, i.e., the best-reinnervated regions, with no obvious side differences. Consistent with previous observations [22], the vast majority (i.e., about 75–80%) of the BrdU-immunoreactive nuclei in the SGZ of all animals also expressed the mature neuron-specific marker NeuN, as seen with double immunofluorescence (Figure 6E–H), and their density was clearly associated with the occurrence of DBH-immunoreactive fiber innervation (Figure 6A–D). Likewise, in no case was any significant group differences detected in the numbers of BrdU-immunoreactive neurons in the hilus proper (Table 3).

## 3. Discussion

The aim of the present study was to investigate whether reinstatement of noradrenergic neurotransmission in the depleted hippocampus with implanted LC-derived neuroblasts—but not its lack, following implantation of control non-noradrenergic tissue—would simultaneously ameliorate lesion-disrupted working memory and neural progenitor proliferation in the SGZ, an issue that has never been addressed before.

### 3.1. Effects of Lesion

Extending recent findings [22,25], selective noradrenergic denervation of the hippocampus with local injections of the anti-DBH-saporin immunotoxin appeared sufficient to severely disrupt animals’ performance in the working memory version (i.e., the RAWM) of the Morris water maze task without seemingly affecting reference memory. These results confirm previous reports of working memory deficits associated with noradrenergic dysfunctions induced by either aging [27,28], reversible LC inactivation [29,30] or non-selective lesioning [31,32,33,34], and strongly suggest an essential role for hippocampal NA in the regulation of cognitive function.

The toxin infusion protocol used here produced robust but territorially defined noradrenergic depletion [24] and appeared suitable to address the specific contribution of hippocampal NA to spatial learning and memory. The same approach was previously used by Steckler and colleagues [35] to selectively remove the septo-hippocampal cholinergic innervation believed to regulate short-term spatial memory. In that study, however, the deficits in an operant delayed response task induced by the lesion were seen as rather modest, suggesting that cholinergic inputs to the hippocampus may play a less critical role, compared to the noradrenergic afferents for modulating working memory. Arguably, an altered hippocampal cholinergic neurotransmission, induced by noradrenergic loss, may have participated in the cognitive impairments reported here. It has been proposed, for example, that NA from the LC exerts a permissive action on acetylcholine release from septo-hippocampal terminals [36]; therefore, the cognitive dysfunctions associated with the loss of hippocampal NA might well result from a decreased cholinergic neurotransmission [29]. In fact, the two transmitter systems have been reported to closely interact, mainly at the hippocampal level, to regulate working memory [36,37,38,39]. In such a scenario, the working memory deficits observed here are likely to result from a concurrent loss of cholinergic neurotransmission unspecifically induced by the hippocampal noradrenergic lesion. Although this possibility cannot completely be ruled out, based on the present results, it seems rather unlikely. First, previous studies have reported increased, rather than reduced, cholinergic activity in the hippocampus of patients with mild cognitive impairment/early AD and NA deficiency [13,40] or rats with neurotoxic NA depletions [41,42]. Second, in a recent study, de Leo et al. [43] observed that selective dual ablation of basal forebrain cholinergic and LC noradrenergic afferents in rats produces no significant worsening of working memory deficits beyond those induced by a noradrenergic depletion alone. Third, the magnitude of the functional recovery promoted by the grafted noradrenergic neuroblasts (discussed below) argues against any significant cholinergic contribution to the functional deficits caused by the noradrenergic lesion.

Clearly, the production of massive (albeit regionally discrete) noradrenergic depletions in developing animals does not seem to best recapitulate age-related and progressive events such as those occurring in AD, and thus further studies with a specific design and analyses at different time points post-lesion, will be necessary to suitably address this issue.

In any event, highly relevant to the concept of an important implication of ascending noradrenergic inputs in various aspects of AD pathogenesis is the observation that hippocampal NA loss is also sufficient to produce marked reductions in SGZ progenitor proliferation, which is in line with the data from recent investigations [19,20,21,22,23]. The present results confirm and extend those findings, reporting a similar magnitude (>60%) in progenitor proliferation decrease in the SGZ of rats with selective hippocampal NA depletions. More interestingly, in the present study, the lesion-induced depression of neurogenesis, as well as the working memory deficits could be detected following a relatively restricted (i.e., hippocampal only, with no apparent cortical involvement) NA loss, and was seen unmodified up to about 9 months post-surgery. The long-term stability of these effects, concurrently produced by the selective lesion, confirms the central role of hippocampal NA in the regulation of both features and further indicates that they may indeed be associated, a finding in line with the strong correlation previously reported between progenitor proliferation in SGZ and RAWM performance [22].

### 3.2. Effects of Transplants

LC-derived neuroblasts implanted into the previously denervated hippocampus of adult rats show excellent capacity for survival and functional integration [44] and have been seen able to restore fairly normal patterns of noradrenergic innervation, transmitter turnover and release [45,46]. Surprisingly, however, little is known about the transplant-promoted reinstatement of noradrenergic innervation and neurotransmission in the hippocampus as a prerequisite to restoring perturbed cognitive abilities in experimental studies. In recent investigations, thanks also to the possibility to selectively and efficiently ablate noradrenergic LC neurons, intrahippocampal grafts of embryonic LC tissue have been reported to significantly ameliorate aspects of spatial cognitive function, namely working memory, in rats with subcomplete NA depletions [24,25]. The present results extend those findings and show that reinnervation of the deafferented hippocampus with implanted noradrenergic neuroblasts is sufficient to significantly ameliorate the lesion-induced working memory impairments, with no apparent effects on other aspects of spatial navigation, such as those related to reference memory, which by contrast may depend upon the integrity of ascending inputs from cholinergic neurons in the basal forebrain [47]. In keeping with previous interpretations [22,43], the LC-derived noradrenergic afferents to the hippocampus may have a more crucial role than the septo-hippocampal cholinergic projections in working memory function, therefore suggesting that the two transmitter systems may be implicated in the regulation of different memory domains.

Of particular interest in the present study is the observation of restorative effects, upon progenitor proliferation in the SGZ, of reinstated noradrenergic inputs from transplanted LC tissue, overall indicating an essential role for hippocampal NA also in the normal functioning of the hippocampal neurogenic niche.

Within such a framework, attempts at increasing/normalizing neurogenesis in the hippocampus of experimental animals have used pharmacological tools, including antidepressants [48] or adrenergic receptor agonists and antagonists [20,21,49,50]. In other studies, an important regulatory role has been described also for the raphe-derived serotonergic inputs [51,52], whose reinstatement, using transplanted neuroblasts in rats with serotonergic depletion, has been reported to result in increased proliferation in the SGZ [53]. Unfortunately, however, the pharmacological manipulations or the lesioning agents adopted in these investigations may lack selectivity. Moreover, in none of these studies were the effects upon cognitive performance specifically investigated, which leaves still relatively uncertain whether hippocampal NA (or serotonin) would simultaneously contribute to the regulation of both hippocampus-dependent spatial working memory and neurogenesis.

The use, in the present study, of a highly selective noradrenergic lesioning tool in conjunction with intracerebral transplantation procedures, well-established hippocampus-dependent swim maze tasks and unbiased stereological estimations of neuronal loss and hippocampal neurogenesis has made it possible to suitably address this issue. Thus, intrahippocampal grafts of NA-rich, but not control cerebellar, tissue were seen to significantly reverse the lesion-induced decrease in the numbers of proliferating progenitors in the SGZ, as estimated using unbiased stereology. This is of importance, as it indicates a specific noradrenergic influence upon progenitor proliferation in the adult rat hippocampus, possibly similar to that exerted by serotonergic [51,52] or cholinergic afferents [54,55,56], which is consistent with the notion of a positive regulatory role upon hippocampal neurogenesis played by multiple neurotransmitter systems [57].

Notably, the transplant-induced increase in progenitor proliferation in SGZ could be detected even many (at least nine) months after lesioning and grafting, a finding which—to the best of our knowledge—has never been reported so far. In fact, the only study addressing the effects of transplant-induced recovery of neurotransmission upon progenitor proliferation in SGZ [53] used much shorter survival times (about 1 month) post-transplantation. Taken together with the above considerations about the long-term stability of the cognitive deficits produced by the lesion, it is therefore plausible that both neurogenesis in the dentate gyrus and hippocampus-dependent spatial working memory share similar mechanisms regulated by noradrenergic inputs ascending from the LC.

## 4. Materials and Methods

### 4.1. Subjects and Experimental Design

A total of 52 equally distributed male and female Sprague-Dawley rats from five different litters (provided by the animal facility at the University of Trieste) maintained in high-efficiency cage units (Tecniplast, Buguggiate, Italy), were used. The pups were randomly allocated into five groups: unoperated controls (Intact, *n* = 6), vehicle-injected (Vehicle, *n* = 6), lesioned controls (Lesioned, *n* = 16), lesioned and transplanted with embryonic LC neurons (LC Transplant, *n* = 16), lesioned and transplanted with control embryonic cerebellar tissue (CBL Transplant, *n* = 8). Post-surgical handling and monitoring of the pups were as previously described [24,25,43]. At about 32 weeks of age, the animals were sequentially evaluated for sensory-motor and spatial navigation abilities after which, about 36 weeks post-lesioning and grafting, the animals were perfused, and the brains were processed for immunohistochemistry and quantitative morphological analyses.

All the experimental procedures were approved by the Ethical Committee at the University of Trieste and closely followed the Italian Guidelines for Animal Care (D.L. 116/92 and 26/2014), the European Communities Council Directives (2010/63/EU) and the National Institutes of Health “Guide for the care and use of Laboratory animals” (8th ed. 2011).

### 4.2. Lesion and Transplantation Surgery

Selective ablation of noradrenergic fibers in the developing hippocampus was performed on 4-day-old (post-natal day, PD, 4) pups under hypothermic anesthesia, as detailed previously [24].

Four days later (on PD8), a noradrenergic-rich and a control cerebellar cell suspension were prepared [24,25,47,58,59] following a modified protocol based on the cell suspension technique [60]. A detailed account of the transplantation procedure can be found in Gulino et al. [24] and Pintus et al. [25]. After each surgery, the animals were locally treated with 2.5% lidocaine-prilocaine cream (EMLA, AstraZeneca, Milano, Italy) and then allowed to reacquire normal body temperature and full recovery, prior to being returned to the mothers and left undisturbed.

### 4.3. Behavioural Tests

Testing sessions took place consistently between 9:00 a.m. and 3:00 p.m. In order to check for non-specific lesion-induced motor deficits, from about 4 weeks post-lesion, all animals were evaluated monthly, using simple tests of gait and coordination [61]. Thus, locomotion and support were assessed after placing the rat onto an 80 cm long wooden ramp connected to the animal’s home cage, which was maintained either horizontally or inclined at a 45° angle. An inclined (75°) 80 × 30 cm framed grid made of coarse mesh chicken wire was also used, where the rats were placed head-down, being scored for their ability to reverse direction and climb onto it.

### 4.4. Morris Water Maze

The Morris Water Maze (MWM) task, originally developed by Morris [62] and known to be sensitive to the effects of hippocampal damage [63], was used as a well-established and efficient tool to assess spatial learning and memory in animals.

Testing was carried out using a circular pool, 150 cm in diameter and 50 cm deep filled to a depth of 35 cm with room temperature (20 °C) water and located in a room containing many external cues for orientation. Four equally spaced points (conventionally indicated as north, south, east and west) divided the tank into four quadrants and served as start locations. In the center of the south–west (SW, training) quadrant, a circular platform (10 cm in diameter) was fixed to the floor of the pool with its top 2 cm below the water surface, onto which the animal could escape. Four annuli were defined as a circular area in the center of each quadrant, where the platform would have been positioned.

The animals became first habituated to the pool environment by receiving a free 60 s swim, then a three-day cued testing was administered, during which the platform was made visible using a striped flag and its position changed randomly on each of four daily trials. Subsequently, two different paradigms, designed to assess reference and working memory abilities, were used.

In the reference memory version of the water maze task, the animals were given four trials a day over seven consecutive days, with a 30 s inter-trial interval, followed by a spatial probe trial, administered after the last trial on day 7, during which the animals were allowed to swim freely for 60 s upon removal of the platform. The latency, distance and swim distribution in the various quadrants during the spatial probe trial were recorded.

Two days after completion of the reference memory test, the animals underwent a five-day assessment of spatial working memory using a radial arm water maze (RAWM) procedure [64]. The apparatus consisted of the same circular pool as above, with six swim alleys (50 cm length × 20 cm wide) radiating out of an open central area, in which the platform was randomly moved to a new arm each day. In this task, the parameters recorded over five daily trials included latency and the errors made upon entering an incorrect arm. Detailed accounts of the procedures are reported elsewhere [22,24,61].

### 4.5. Injections of 5-Bromo-2′Deoxyuridine and Post-Mortem Procedures

Upon conclusion of the behavioral testing, about 36 weeks post-lesioning and grafting, all animals received a single intraperitoneal (i.p.) injection of the thymidine analog 5-bromo-2′deoxyuridine (BrdU, 200 mg/kg. Sigma) dissolved in sterile PBS. Two hours later, the rats were deeply anesthetized (sodium pentobarbital, 6 mg/100 g i.p.) and perfused via the ascending aorta with 50–100 mL of room temperature saline followed by 250–300 mL ice-cold 4% paraformaldehyde in PBS (pH 7.4). The brains were rapidly removed and postfixed for 2 h, and then the tissue was processed as reported previously [22,24,25,59]. For the subsequent morphometric determinations, DBH or BrdU immunohistochemistry as well as BrdU and neuron-specific nuclear protein (NeuN) double immunofluorescence were conducted following previously reported protocols [22,54,56].

### 4.6. Microscopic Analysis and Quantitative Evaluation

Analyses were conducted on coded slides by unaware investigators. For stereological assessments, the optical fractionator principle [65] was followed to obtain unbiased estimates of the numbers of noradrenergic neuronal populations within the exact boundaries of the LC/SubC regions [1,66], as previously reported [22,24,25].

Stereological quantification of BrdU-immunoreactive cells within the SGZ and hilus of the hippocampal DG were carried out bilaterally on 5 to 7 coronal sections per rat, located between 3.0 and 4.5 mm caudal to bregma, thus encompassing the dorsal hippocampus, as previously described [22,54,56].

Changes in the density of hippocampal noradrenergic innervation, possibly induced by either the immunotoxin treatment or grafted cells were assessed with densitometry, using the Image 1.61 image analysis software [67], as detailed in previous studies [22,25].

### 4.7. Statistical Analysis

Data fulfilled the criteria for normal distribution and were therefore analyzed using parametric tests for all statistical comparisons. Group differences in behavioral performance as well as in cell numbers or innervation density were assessed using either one-way analysis of variance (ANOVA) or two-way mixed ANOVA, as appropriate, followed by Tukey HSD post hoc test. Data are presented as means ± standard error of the mean (sem), and differences were considered significant at *p* < 0.05.

## 5. Conclusions

The present data suggest that the same noradrenergic influence arising from the LC is capable of providing a positive regulation to seemingly distinct functional domains, namely hippocampus-dependent spatial working memory and proliferation of newly generated progenitor in the SGZ of the adult rat. If so, then NA loss and altered neurogenesis in the hippocampus are to be considered as early and concurrent events contributing to cognitive decline. Identification of novel approaches capable of increasing hippocampal noradrenergic neurotransmission and neurogenesis may thus prove useful for the prevention and treatment of the memory decline associated with AD.

## Figures and Tables

**Figure 1 ijms-24-05613-f001:**
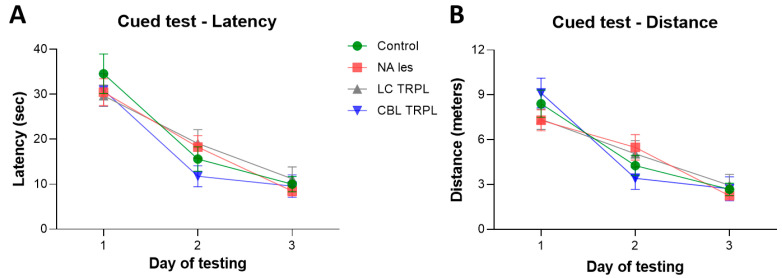
Performance in the cued version of the Morris Water Maze test. All animals in the Control, Lesioned (NA less), LC-transplanted (LC TRPL) and CBL-transplanted (CBL TRPL) groups, exhibited similar escape latencies (**A**) and swim distances (**B**). Each point represents the mean value ± SEM for the block of four trials administered each day.

**Figure 2 ijms-24-05613-f002:**
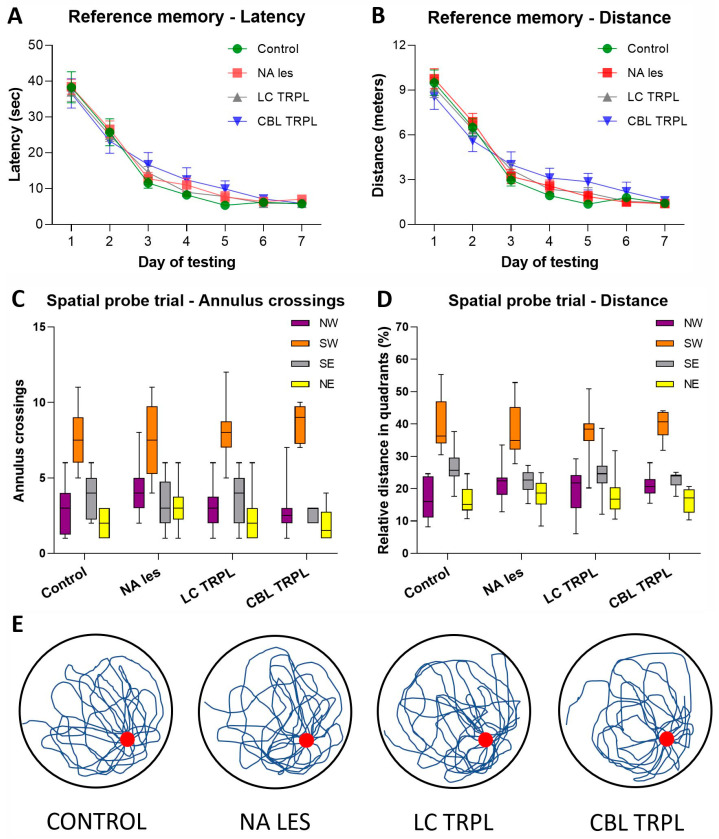
Morris Water Maze reference memory test performed about 32 weeks post-lesioning and grafting. All animals showed equally efficient abilities both in latency (**A**) and distance (**B**). Each sample point represents the mean value for the block of four trials on each of seven consecutive days of testing ± SEM. In the lower diagrams, the average number of annulus crossings (**C**), the mean relative distance (**D**) swam in each quadrant during the spatial probe trial and the actual swim paths taken by representative rats from the various groups (**E**) are illustrated. All animals mainly swam in the training (SW) quadrant, indicating that the selective noradrenergic lesion or the grafting procedures have no effects on reference memory performance.

**Figure 3 ijms-24-05613-f003:**
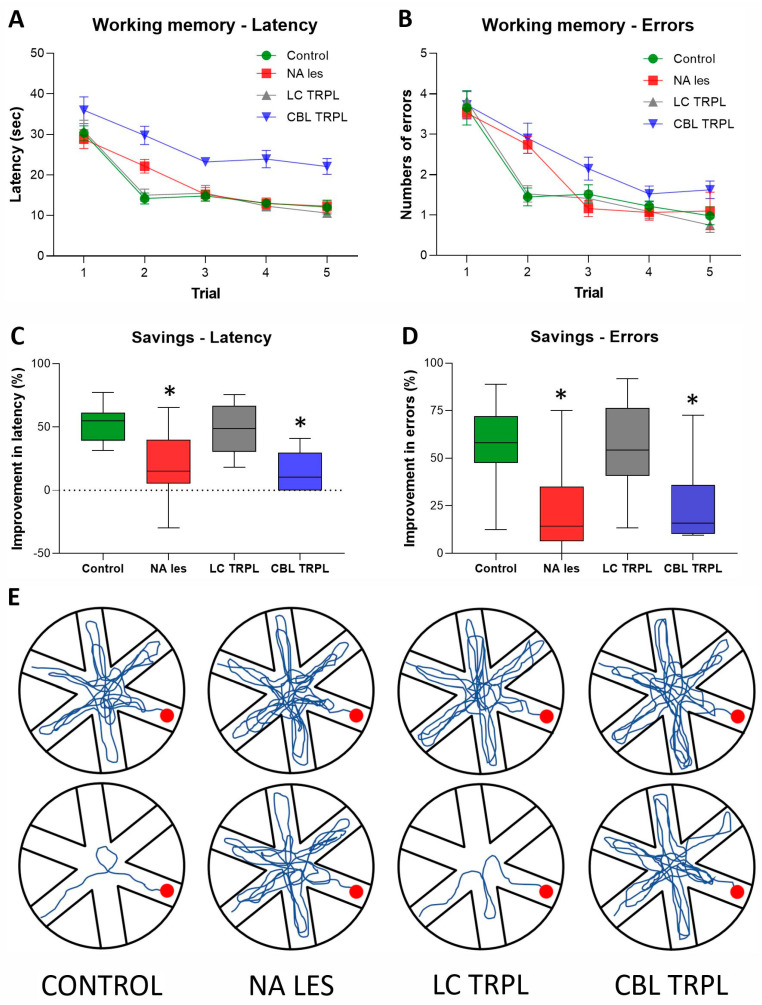
Working memory performance in the Radial Arm Water Maze task performed about 34 weeks post-lesioning and grafting. The latency (**A**) and the number of entry errors (**B**) required by the animals to locate the hidden platform are illustrated. Sample points represent the mean latency and errors ± SEM recorded during each of 60 s trials over five consecutive testing days. In the lower diagrams, performances are plotted as percentage improvement (savings) between trials 1 and 2 for latency (**C**) and errors (**D**). In (**E**), the actual swim paths taken by representative rats from the different groups are illustrated. Note the marked impairments in the Lesioned and CBL-transplanted animals, and the near-normal performance of LC-transplanted animals, with respect to Control. The asterisk indicates a significant difference between the Control and LC-Transplanted groups at *p* < 0.01.

**Figure 4 ijms-24-05613-f004:**
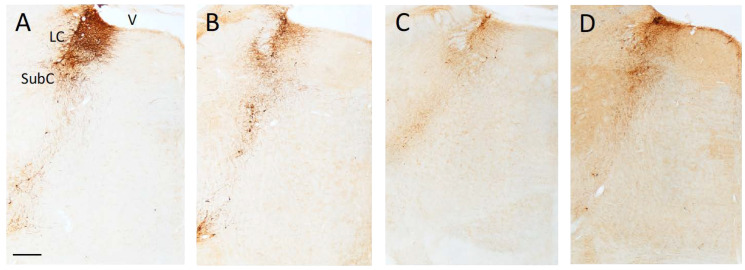
Representative photomicrographs of DBH-immunostained coronal sections illustrating the distribution of noradrenergic neurons in the LC and SubC nuclei of Control (**A**), Lesioned (**B**), LC-transplanted (**C**) and CBL-transplanted animals (**D**). Note in B-D, the lesion-induced loss of immunoreactive neurons and proximal processes was fairly complete in the dorsalmost part of the nucleus, with some sparing in the more ventral SubC region. Scale bar in A: 500 µm. LC, Locus Coeruleus; SubC, SubCoeruleus complex; V, Ventricle.

**Figure 5 ijms-24-05613-f005:**
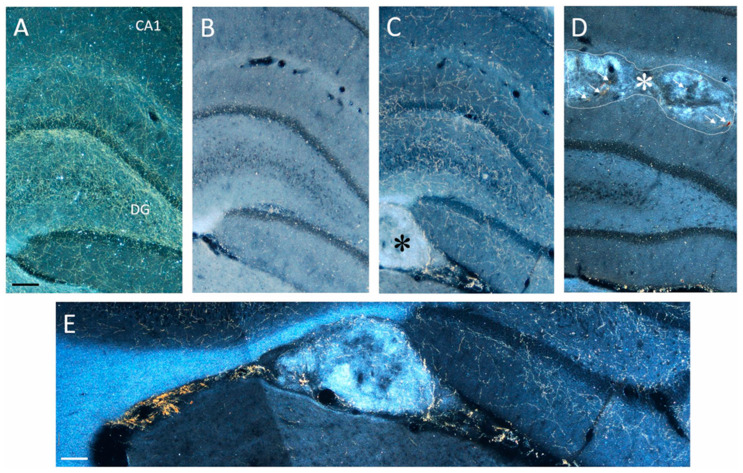
Darkfield photomicrographs illustrating the distribution of DBH-immunoreactive fibers in the hippocampus of Control (**A**), Lesioned (**B**) LC-transplanted (**C**) and CBL-transplanted animals (**D**) about 36 weeks after lesioning and grafting. Note the near complete loss of DBH-positive fibers in the hippocampus of Lesioned animals (**B**) and CBL-transplanted animals (**D**) compared to control (**A**) and the close to normal pattern of noradrenergic innervation reinstated by the transplanted LC tissue (**C**). In (**E**), the morphology of the transplanted LC tissue and the outgrowing DBH-immunoreactive fibers is illustrated at higher magnification. Note the remarkable anatomical integration of the LC graft within the host tissue environment and the smooth fiber growth at the graft-host border. CA, cornu ammonis of the hippocampus; DG, dentate gyrus. The asterisks in (**C**,**D**) indicate the location of the LC and CBL graft deposits, respectively. The dashed line and the arrows in (**D**) indicate, respectively, the boundaries of the CBL tissue graft and the occasional presence of DBH-immunoreactive elements within it. Scale bar in (**A**): 100 µm; in (**E**): 50 µm.

**Figure 6 ijms-24-05613-f006:**
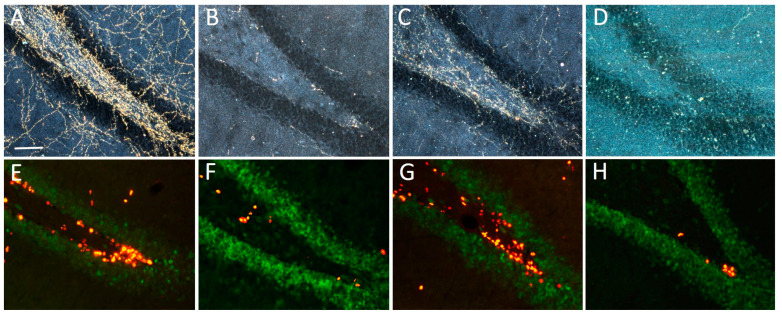
Representative micrographs illustrating DBH-positive innervation (**A**–**D**) or BrdU (red) and NeuN (green) immunoreactive neurons (**E**–**H**) in the hippocampal dentate gyrus about 36 weeks after lesioning and grafting. Note the dramatic loss of DBH-immunoreactive fibers and BrdU/NeuN positive progenitors in the Lesioned (**B**,**F**) and CBL-transplanted animals (**D**,**H**), compared to Control (**A**,**E**), which by contrast appeared reinstated in LC-transplanted animals (**C**,**G**). Scale bar in (**A**): 50 µm.

**Table 1 ijms-24-05613-t001:** Motor performance.

Group	Equilibrium Time on Ramp (%)	Latency to Cross Ramp (s)	Latency to Reverse on Grids (s)	Number of Falls in Grids
Control (*n* = 12)	96.9 ± 0.7	6.9 ± 0.4	6.2 ± 0.7	2.4 ± 0.5
Lesioned (*n* = 16)	97.8 ± 0.6	6.9 ± 0.3	6.6 ± 0.7	2.5 ± 0.5
LC Transplant (*n* = 16)	98.2 ± 0.5	7.0 ± 0.3	6.3 ± 0.5	2.4 ± 0.5
CBL Transplant (*n* = 8)	96.3 ± 1.3	7.1 ± 0.5	7.0 ± 0.9	2.4 ± 0.7

Motor tests evaluated postural and locomotive form onto an 80 cm long wooden ramp or an inclined (75°) grid, both connected to the animals’ home cage. Values represent the average of four determinations ± SEM.

**Table 2 ijms-24-05613-t002:** Stereological and densitometric estimates of DBH-immunoreactive neurons and fibers in the Locus Coeruleus and the various hippocampal terminal fields, respectively.

Group	DBH-ir Neurons in LC/SubC	DBH-ir Fibers in CA1	DBH-ir Fibers in CA3	DBH-ir Fibers in DG
Control (*n* = 12)	1782.4 ± 44.2	62.0 ± 3.6	61.6 ± 1.4	64.3 ± 2.5
Lesioned (*n* = 16)	395.6 ± 36.1 *	18.9 ± 1.1 *	21.0 ± 1.3 *	20.3 ± 0.8 *
LC Transplant (*n* = 16)	458.6 ± 48.8 *	55.4 ± 1.8	57.3 ± 2.8	59.3 ± 3.5
CBL Transplant (*n* = 8)	451.7 ± 23.6 *	23.6 ± 0.4 *	21.4 ± 0.8 *	21.5 ± 1.2 *

Values indicate the estimated total number of DBH-immunoreactive neuronal profiles in the LC, and the relative density scores (±SEM) of the DBH-positive innervation in each of the various subdivisions of the hippocampal formation, i.e., Cornu Ammonis (CA) 1 and 3 and the Dentate Gyrus (DG). Asterisks indicate a significant difference from Normal (*p* < 0.01).

**Table 3 ijms-24-05613-t003:** BrdU-immunoreactive cell counts.

Group	BrdU-ir Cells in SGZ	BrdU-ir Cells in Hilus
Control (*n* = 12)	8480.5 ± 607.9	2675.7 ± 741.0
Lesioned (*n* = 16)	3242.5 ± 310.6 *	2018.1 ± 243.9
LC Transplant (*n* = 16)	9047.3 ± 575.3	2358.5 ± 481.2
CBL Transplant (*n* = 8)	3018.1 ± 233.8 *	2838.6 ± 289.7

Values represent the total number of BrdU-immunoreactive cells per bilateral SGZ and hilus, as estimated using stereology. Asterisks indicate significant differences between Control and LC-Transplanted groups (*p* < 0.01).

## Data Availability

The data presented in this study are available on request from the corresponding author.

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
