# Peer review of "Hippocampal Noradrenaline Is a Positive Regulator of Spatial Working Memory and Neurogenesis in the Rat"

_ijms, 2023, doi:10.3390/ijms24065613_

Round 1

Reviewer 1 Report

In this study, the authors reported the investigation of the possible relationships between NA-regulated neurogenesis and hippocampus-dependent cognitive abilities by transplanted LC-derived neuroblasts. The results clearly show that hippocampal noradrenaline is a positive regulator of spatial working memory and neurogenesis. The manuscript is well-written and organized. The conclusions are supported by the data. Minor revision to improve the quality of this manuscript is needed before acceptance for publication.

Comments:

1.      In the result section, I suggest explaining the purpose of each experiment in each sub-section more clearly.

2.      Please describe the full names when it appears at the first time. There seems to be no description for “TRPL” throughout the text. The same for Line 130, describe the full name of RAWM.

3.      Table 2, the LC TRPL and CBL TRPL show similar number of DBH-immunoreactive neurons in the LC/SubC and it is significantly less compared with the Control group. Please explain this difference.

4.      Can the authors discuss the limitations and future perspectives of this study?

Author Response

Referee 1

In this study, the authors reported the investigation of the possible relationships between NA-regulated neurogenesis and hippocampus-dependent cognitive abilities by transplanted LC-derived neuroblasts. The results clearly show that hippocampal noradrenaline is a positive regulator of spatial working memory and neurogenesis. The manuscript is well-written and organized. The conclusions are supported by the data. Minor revision to improve the quality of this manuscript is needed before acceptance for publication.

Comments:

  1. In the result section, I suggest explaining the purpose of each experiment in each sub-section more clearly. This has now been done for the behavioural testing, see lines 115-116 and 141-142 in the revised submission

  1. Please describe the full names when it appears at the first time. There seems to be no description for “TRPL” throughout the text. The same for Line 130, describe the full name of RAWM. This has now been done for ‘Cerebellar’ (‘CBL’, line 75), the group names in the figures (caption for Figure 1, lines 106-107) and for ‘Radial arm Water Maze’ (‘RAWM’ line 141). ‘TRPL’ appears for the first time in Table 1. In this table and in Tables 2-3 it has been indicated as ‘Transplant’.
  2. Table 2, the LC TRPL and CBL TRPL show similar number of DBH-immunoreactive neurons in the LC/SubC and it is significantly less compared with the Control group. Please explain this difference. A significantly reduced number of DBH-immunoreactive neurons in the LC/SubC of all lesioned animals (whether transplanted or not) compared to Control is an expected (and even a most welcome) finding, as it indicates the observed reinnervation in the target territory (i.e. the hippocampus) is almost totally graft-derived and not the result of an incomplete lesion. This is now briefly mentioned in the text (results, lines 214-216).
  3. Can the authors discuss the limitations and future perspectives of this study? This has now been done (Discussion, lines 431-435).

Reviewer 2 Report

Review of manuscript no. ijms-2230255 which has been submitted to International Journal of Molecular Sciences-Manuscript

I want to start with the fact that I am not a specialist in the field of this article, but I will make this review from the structural point of view and the relevance of the subject. In the current context of the study topic, the article entitled “Hippocampal Noradrenaline is a Positive Regulator of Spatial Working Memory and Neurogenesis in the Rat” is very interesting and the theme is well chosen. From my point of view, the obtained results are impressive and initiate new hypotheses in studying central nervous system physiological functions. As a result of the methodology used, a lot of data were obtained which were linked together very efficiently and clearly.

I gladly recommend the article be accepted for publication.

Anyway, below I made some suggestions:

·         Page 9, lines 277-279; Please reformulate for a better understanding of the sentence “The main aim of the present study was to examine the importance of the coeruleo-277 hippocampal noradrenergic projections in the regulation of aspects of spatial learning and 278 memory and, concurrently, of the complex events leading to neural progenitor prolifera-279 tion in the SGZ.”.

Author Response

Referee 2

I want to start with the fact that I am not a specialist in the field of this article, but I will make this review from the structural point of view and the relevance of the subject. In the current context of the study topic, the article entitled “Hippocampal Noradrenaline is a Positive Regulator of Spatial Working Memory and Neurogenesis in the Rat” is very interesting and the theme is well chosen. From my point of view, the obtained results are impressive and initiate new hypotheses in studying central nervous system physiological functions. As a result of the methodology used, a lot of data were obtained which were linked together very efficiently and clearly.

I gladly recommend the article be accepted for publication.

Anyway, below I made some suggestions:

  • Page 9, lines 277-279; Please reformulate for a better understanding of the sentence “The main aim of the present study was to examine the importance of the coeruleo-277 hippocampal noradrenergic projections in the regulation of aspects of spatial learning and 278 memory and, concurrently, of the complex events leading to neural progenitor prolifera-279 tion in the SGZ.”. The sentence has now been rephrased (Discussion lines 354-356, in the revised submission).

Reviewer 3 Report

This is a very interesting noble study in the emerging field of neuroscience, which has potential therapeutic targets in the field. The study is well-designed with appropriate experiments. The results interpretation is sound with appropriate language and claims. I have only a few substantive questions, which are unlikely to affect a decision to accept, and then a number of suggestions on the presentation that I leave to the authors' discretion.

Alzheimer’s disease and cognitive decline are associated with aging. Authors have worked on relatively young/adult mice to study noradrenaline (NA). Do you have experience or literature on the role of NA in old/aging mice?

 Introduction can be improved with NA in naturally occurring cognitive decline and AD.

The p-value and standard deviation can be included in the figures.

Author Response

Referee 3

This is a very interesting noble study in the emerging field of neuroscience, which has potential therapeutic targets in the field. The study is well-designed with appropriate experiments. The results interpretation is sound with appropriate language and claims. I have only a few substantive questions, which are unlikely to affect a decision to accept, and then a number of suggestions on the presentation that I leave to the authors' discretion.

Alzheimer’s disease and cognitive decline are associated with aging. Authors have worked on relatively young/adult mice to study noradrenaline (NA). Do you have experience or literature on the role of NA in old/aging mice? In our study we sought to specifically investigate the noradrenergic contribution to hippocampus-dependent spatial memory and neurogenesis and their possible concurrent reinstatement following NA replacement by transplanted neuroblasts. To this aim we have employed immature animals whose intrinsic developmental plasticity would possibly offer a better milieu for the development of the transplanted tissue. Certainly, producing massive (although discrete) lesions in developing animals may not be seen as an optimal recapitulation of the progressive and age-related events in AD. On the other hand, this would allow to avoid possibly confounding factors such as those related to a general age-dependent failure of several neural systems, whose analysis would require longitudinal studies with a specific design. This is now briefly mentioned in the discussion section (lines 431-435). Although, in the present study, we did not specifically address neurochemical and functional changes in aged animals, other studies did and they are reported in the text (e.g. refs 27 and 28).

Introduction can be improved with NA in naturally occurring cognitive decline and AD. These highly relevant informations are already present in the Introduction (line 41) and adequately referenced both for AD-related cognitive decline (refs 5-15) and impaired neurogenesis (16-18).

The p-value and standard deviation can be included in the figures. P-values are indicated in the captions of figures (e.g. Fig. 3) and Tables (e.g. Tables 2 and 3) where statistically significant differences were to be highlighted. We decided not to indicate them in the case of no significant group differences, leaving the description to the caption or the main text. Where relevant, data were presented as means ± SEM